| Food Microbiology | Research Article

# *Lysinibacillus macroides* 38352 isolated from traditional Chinese fermented foods: a dual effect on ochratoxin A detoxification and immune suppression alleviation

Jiaqing Wang,[1] Haoyuan Duan,[1] Wenzhi Zhang,[1] Hai Li,[1] Zaixing Yang,[1] Chuankun Zhang,[1] Shuhe Zhang,[1] Junjie Guo,[1] Junwei Ge,[1] Fang Wang,[2] Mingchun Gao[1]

**ABSTRACT**  Ochratoxin A (OTA), a nephrotoxic, immunosuppressive, and potentially carcinogenic mycotoxin produced by *Aspergillus* and *Penicillium* species, poses a persistent threat to global food safety and livestock health. Although current biological detoxification strategies can degrade OTA, they often overlook its detrimental impact on vaccine efficacy, and safe probiotic candidates with dual functions of detoxification and immune restoration remain scarce. In this study, we isolated 3 OTA-degrading bacterial strains from 28 types of traditional Chinese fermented foods. Among them, *Lysinibacillus macroides* 38352 exhibited the highest degradation efficiency, removing 51.4% of 2.5 µg/mL OTA within 72 h at 37°C, with further increases observed over time. Mechanistic analysis revealed that the degradation was mediated not by cell surface adsorption but by secreted metabolites generated through active cellular metabolism. The strain also demonstrated excellent probiotic properties, including tolerance to pH 2–12 and 0.2% bile salts, sensitivity to 15 antibiotics (indicating no resistance), and no pathogenicity in mice. Notably, oral administration of *L. macroides* 38352 significantly reversed OTA-induced immunosuppression, enhancing vaccine-induced specific IgG, neutralizing antibody titers, and cytokine levels (interleukin [IL]-1β, IL-4, IL-12, IL-17, and interferon gamma), while also reducing inflammatory damage and improving growth performance in OTA-exposed broilers. Our study demonstrates that *L. macroides* 38352 is a dual-function probiotic capable of both OTA biodegradation and immune restoration. These findings provide a mechanistically distinct microbe-based strategy that couples mycotoxin detoxification with host immune restoration, advancing microbiome-informed interventions for sustainable livestock production and food safety.

**IMPORTANCE**  Ochratoxin A (OTA) is a widespread food and feed contaminant known for its toxic, carcinogenic, and immune-disrupting effects, which pose serious challenges to animal health and food security. However, existing detoxification approaches rarely address the immune damage caused by OTA, and safe microbial solutions with both detoxifying and immune-supporting capabilities remain limited. This study introduces *Lysinibacillus macroides* 38352, a newly identified probiotic from traditional Chinese fermented food, as a dual-function candidate that not only degrades OTA efficiently through secreted metabolites but also helps repair immune function suppressed by OTA exposure. The strain shows excellent safety features and effectively improves vaccine responses and overall health in poultry. These findings highlight a novel microbe-based strategy that integrates mycotoxin detoxification with immune restoration, advancing the development of functional probiotics for microbiome-driven interventions in sustainable animal production and food safety.

**Peer Reviewers** Nurul Hawa Ahmad, Universiti Putra Malaysia, Selangor, Malaysia; Biao Tang, University of the Chinese Academy of Sciences, Hangzhou, China

Address correspondence to Fang Wang, wangfang06@caas.cn, or Mingchun Gao, gaomingchun@neau.edu.cn.

Jiaqing Wang and Haoyuan Duan contributed equally to this article. The author order was determined by mutual agreement among the authors.

The authors declare no conflict of interest.

See the funding table on p. 18.

**KEYWORDS** OTA, biodegradation, *Lysinibacillus macroides* 38352, broiler, vaccine efficacy

Ochratoxin A (OTA) is a mycotoxin produced by *Aspergillus* and *Penicillium* species, frequently contaminating cereals, animal feed, and fermented foods (1–3). OTA exhibits nephrotoxic, hepatotoxic, and carcinogenic properties (classified as Group 2B by the International Agency for Research on Cancer) and binds strongly to serum albumin, resulting in prolonged accumulation in the body with a half-life of approximately 35 days (4, 5). OTA exposure can suppress vaccine-induced antibody responses, leading to vaccination failures in livestock and posing serious challenges to disease control efforts (6). The global livestock industry suffers substantial annual economic losses due to OTA contamination, and current food processing technologies are largely ineffective at eliminating the toxin (7, 8). Despite stringent regulations, such as the European Union's limit of 100 µg/kg for poultry feed (EC 1881/2006), OTA contamination remains widespread (9). Therefore, there is an urgent need to develop effective and safe detoxification strategies that not only eliminate OTA and minimize its immunotoxic effects but also mitigate its interference with vaccine-induced immune responses.

Microbial agents, including *Bacillus* and lactic acid bacteria, as well as enzymes like carboxypeptidase, have been explored for OTA degradation due to their environmental friendliness and efficiency (10). For instance, *Aspergillus oryzae* demonstrated the potential of fungal resources by degrading 94% of OTA within 72 h through the secretion of amidase (11). *Lactobacillus brevis* adsorbs OTA via cell wall peptidoglycan, though the potential re-release of toxins in the intestine limits its application (10). A novel amide hydrolase, ADH3, can degrade 50 µg/L OTA within 90 seconds; however, its poor thermal stability (losing over 50% activity below 37°C) hinders industrial application (12). Additionally, *Bacillus* CotA laccase has been shown to alleviate aflatoxin B1 (AFB1)-induced intestinal barrier damage by increasing the villus height/crypt depth ratio and upregulating the mRNA expression of tight junction proteins TJP1 and ZO-1, while reducing the expression of inflammation-related genes in the jejunum (13). Despite advancements in degradation technologies, the immunosuppressive effects of OTA have not been systematically studied, and there is a lack of strains capable of both detoxification and immune restoration.

However, most gut-derived microorganisms exhibit OTA degradation rates below 30%; for example, *Saccharomyces cerevisiae* typically degrades only about 8%–12% of OTA under animal intestinal conditions (14, 15). Although some strains, like lactic acid bacteria, are considered probiotics, many OTA-degrading microbes lack comprehensive *in vivo* safety evaluations, and their long-term use may disrupt host gut microbiota balance or induce inflammatory responses (16, 17). There is a pressing need for multifunctional strains that combine high detoxification efficiency (>50%), *in vivo* safety (non-pathogenicity), and immune modulation (enhancing vaccine efficacy). Traditional fermented foods, known for their microbial diversity and long-standing safety, are ideal sources for screening multifunctional strains (18–21). Their symbiotic microorganisms, including lactic acid bacteria and bifidobacteria, produce metabolites with potential immunomodulatory and detoxifying properties (22–24). Therefore, this study hypothesizes that strains with both OTA-degrading and immune-restorative functions can be isolated from traditional Chinese fermented foods.

The present study aimed to screen multifunctional strains and systematically evaluate their detoxification efficacy, safety, and ability to restore vaccine-induced immunity. The screening criteria focus on tolerance to OTA (50 µg/mL), extreme pH (2–12), and bile salts (0.2%), combined with *in vitro* probiotic characteristics such as non-hemolytic activity and antibiotic susceptibility. Furthermore, the OTA degradation activity of *Lysinibacillus macroides* 38352 (*L. macroides* 38352) was assessed using high-performance liquid chromatography (HPLC) to analyze its supernatant, intracellular components, live cells, and dead cells, elucidating its mechanism of action. Also, the relationship between OTA degradation capacity and factors such as time and cell concentration was evaluated.

Finally, the strain's ability to counteract OTA-induced immunosuppression was validated in a broiler chicken model by measuring antibody titers, cytokine levels, and intestinal pathology. These findings suggest that *L. macroides* 38352 could serve as a feed additive to simultaneously address OTA contamination and vaccine failure, offering an innovative solution for sustainable livestock farming.

## MATERIALS AND METHODS

### Materials and culture conditions

OTA and other chemical reagents used in this study are of analytical grade. The OTA standard was purchased from Sigma (St. Louis, MO, USA) and was prepared in HPLC-grade methanol.

In this study, 28 samples of traditional Chinese fermented foods, including bean drum, preserved beancurd, leek flower, pickled cabbage, shrimp sauce, tofu milk, fermented glutinous rice, and sour soup noodles, were collected during 2021, transported at low temperature, and stored at 4°C. They were collected from different cities in China, including Heilongjiang, Inner Mongolia, Hebei, Guizhou, Gansu, Guangdong, Shandong, and Liaoning provinces.

The sample (1 g) was mixed with sterile normal saline (9 mL). The supernatant (100 µL) was spread over Luria–Bertani (LB) agar and incubated at 37°C for 12 h. Colonies exhibiting distinct morphologies (based on size, shape, color, and margin) were selected as suspected strains and subjected to purification and preservation.

Single colonies of strains isolated above were inoculated in screening medium containing different OTA concentrations (2.5, 10, 25, 50 µg/mL) and incubated at 37°C in warm boxes for 7 days.

Well-growing colonies were selected according to the turbidity of the bacterial solution and kept in LB agar medium. Primary screening medium: $(NH_4)_2SO_4$ 5.0 g, $KH_2PO_4$ 2.5 g, $MgSO_4$ 1.0 g, $Na_2HPO_4 \cdot 12H_2O$ 0.5 g, $CaCl_2$ 0.1 g, 20 g agar, distilled water 1 L, pH 6.5, 121°C sterilized for 15 min. A filter-sterilized OTA solution was added.

*Clostridium perfringens* type A (C57-8) was purchased from the China Veterinary Drug Administration. A total of 20 female KM mice (SPF class) weighing 20 ± 2 g were used for experimental purposes. Mice were purchased from Liaoning Changsheng Biotechnology Company (Liaoning, China). The mice were kept under specific pathogen-free conditions and had free access to rodent food and tap water during a 12-h cycle of light and darkness. A total of 210 one-day-old male broiler chicks were obtained from Liaodian Hongyan Breeding Farm, Acheng District, Harbin, China, and housed in the animal facility of Northeast Agricultural University. The Ethical Committee of the Institute approved all scientific experiments. All applicable international and national guidelines for the care and use of animals in experiments were followed and approved by the Institutional Committee of Northeast Agricultural University (NEAUEC20210326).

### Determination of OTA by using HPLC

OTA determination by using HPLC was performed according to the method of reference 25. Following the incubation period, microbial cells were separated through centrifugation at 8,000 rpm for 10 min. The resulting supernatants were subsequently filtered using 0.22 µm pore size filters and subjected to analysis employing HPLC. The HPLC analysis was carried out utilizing an Agilent Technologies instrument (1260II Prime), equipped with a fluorescence detector set to excitation and emission wavelengths of 333 nm and 460 nm, respectively. The analysis employed a reversed-phase Agilent Eclipse Plus C18 column (4.6 × 150 mm, 5 µm) with an injection volume of 20 µL. The mobile phase was water/acetonitrile/methanol (50:35:15, vol/vol/vol) at a flow rate of 1.0 mL/min.

## *In vitro* probiotic characteristics of bacterial strains

According to the method of reference 26, the probiotic properties of the isolated strains, such as tolerance to strong acid and alkali, resistance to bile salts, hemolytic properties, and susceptibility to antibiotics, were analyzed.

### Acid and alkali tolerance

After centrifuging 1 mL of fresh bacterial solution ($1 \times 10^8$ CFU/mL) of overnight culture at 3,000 rpm for 2 min, it was resuspended in LB with pH 1, 2, 12, and 7.4 (control) and cultivated for 2 h at 37°C. It was then rinsed three times with phosphate-buffered saline (PBS) (pH = 7.4). After that, 200 µL of cell suspension was diluted, spread out on an LB agar plate, and cultivated for 12 h at 37°C. The tolerant strains were screened, and the overall viable numbers were noted.

### Bile salt tolerance

Bile salt tolerance assay was performed according to the method described by Huinan Chen et al. with modifications (26). Briefly, the fresh bacterial culture was centrifuged at 3,000 rpm for 2 min. The supernatant was discarded, and the bacterial pellet was collected. The pellet was resuspended in LB liquid medium containing 0.2% (wt/vol) bile salts, thoroughly mixed by pipetting, and incubated at 37°C for 2 h. After another centrifugation step to remove the supernatant, the bacteria were resuspended in an equal volume of sterile PBS buffer. To evaluate bacterial survivability, the number of viable cells was quantified. The resulting suspension was serially diluted 10-fold in sterile PBS. From each dilution, 10 µL was spotted onto LB agar plates, with three replicates per dilution. The plates were incubated at 37°C for 24 h, and colonies were counted. The number of viable bacteria was expressed as colony-forming units per milliliter (CFU/mL).

### Hemolytic properties

Hemolytic activity was determined on sheep blood agar plates (Biocell BioTech., Co., Ltd., Zhengzhou, China). Briefly, 200 µL of the bacterial culture was first spread onto LB agar plates and incubated at 37°C for 24 h to obtain fresh colonies. Using a sterile pipette tip, a single colony was picked and gently spotted onto the surface of a sheep blood agar plate. The plates were incubated at 37°C for 24 h and then examined for the presence of a clear zone (zone of hemolysis) around the bacterial growth. The diameter of the hemolytic zone was measured using a caliper. A positive control and a negative control were included in each assay. After that, the observation continued for a week at 4°C. The entire experiment was independently conducted on three separate days to ensure reproducibility (biological triplicates).

### Antibiotic susceptibility

Antibiotic susceptibility was assessed using standard commercial discs (KONT Biology & Technology Co., Ltd., Wenzhou, China), each containing a predefined volume of antibiotic solution as specified by the manufacturer. In particular, Mueller-Hinton agar plates were equally covered with the isolated bacteria ($1 \times 10^8$ CFU/mL), and the antibiotic sensitivity was measured at 37°C using 30 drug-sensitive discs. To verify the strains' susceptibility, the size of the inhibitory zone (mm) was measured and noted after 12 h of culture. Inhibition zone diameters were interpreted as follows: <8 mm = resistant (R), 9–15 mm = intermediate (I), and >16 mm = susceptible (S). The antibiotic susceptibility was performed according to the manufacturer and the Clinical and Laboratory Standards Institute guidelines (CLSI M02-A13 and M100-ED32).

## Identification of the OTA-degrading strains

The morphological observation of the strains with high OTA-degrading ability was carried out. The identified bacterial strains were assayed by a method described by

Huinan Chen (26). The 16S rRNA gene sequence was amplified by using primer set 27F (5′-AGA GTT TGA TCC TGG CTC AG-3′)–1492R (5′-TAC GGC TAC CTT GTT ACG ACT T-3′), while the *gyrB* gene sequences were amplified using primers UP-1 and UP-2.

The obtained sequences were compared with the GenBank database using BLAST on the National Center for Biotechnology Information (NCBI; http://www.ncbi.nlm.nih.gov). The phylogenetic tree was constructed by the Neighbor-Joining method using the MEGAX program.

## Degradation properties of OTA-degrading strains

To assess the growth patterns of OTA-degrading strains under varying conditions, a temperature range (15°C, 25°C, 30°C, and 35°C) was explored over 48 h, and different time intervals (8, 16, 32, and 48 h) were examined at 37°C. Optical densities (ODs) were measured and recorded at 600 nm. Subsequently, bacterial numbers were calculated using the equation correlating plate counts with OD values and expressed as $\log_{10}$ CFU/mL. To investigate the impact of cell concentration, strains were initially inoculated in LB liquid medium and cultured at 37°C until the broth reached an OD of 1. Cell concentrations were then adjusted to OD 0.8, 0.6, and 0.4 through gradient dilution. Cells with varying dilutions were introduced into 50 mL LB medium. Culture systems with different cell concentrations were agitated at 150 rotations per minute for 48 h at 37°C to assess the effect of cell densities. The bacterial solution was inoculated into LB liquid medium at a 1% ratio, and samples were collected at 2-h intervals. Absorbance was measured at 600 nm using a spectrophotometer to construct the growth curve.

## Active ingredient of OTA-degrading strains

To identify the functional detoxification component, we examined the influence of the culture supernatant, intracellular components, bacterial suspension, and dead cells of isolated strains on OTA removal capacity. The analysis was conducted following the procedure outlined in Yue et al. (27), with certain adjustments made to the methodology. The fresh bacterial suspension was centrifuged at 12,000 r/min for 15 min at 4°C, yielding supernatant and cells. The cells were washed thrice with PBS and then resuspended in 50 mL of PBS, resulting in a bacterial suspension with a concentration of $10^7$ CFU/mL. This suspension was subjected to low-temperature ultrasonication for 15 min. Subsequently, the suspension was centrifuged at 12,000 r/min for 10 min, and the supernatant was collected and filtered through a 0.22 µm membrane to obtain the intracellular fluid of the bacterial strain. The cells, washed thrice, were boiled for 20 min, then resuspended in an equal volume of PBS, serving as dead cells for OTA degradation. The supernatant, bacterial suspension, intracellular fluid, and dead cells were each mixed with OTA solution (100 mL, mass concentration 10 µg/mL). After incubating at 37°C in the dark for 48 h, the culture supernatants were filtered using a 0.22 µm filter membrane. The residual amount of OTA was then determined using HPLC to calculate the degradation rate. Each group was set up with three parallel controls. The blank control for the supernatant group was sterile LB liquid medium, while the blank controls for the bacterial suspension group, intracellular fluid group, and dead cell group were sterile water.

## Safety test in mice

Following the method outlined by Chen et al. with certain modifications (26), a challenge test was conducted to assess the pathogenicity of the three selected strains. A total of 20 female BALB/c mice weighing 20 ± 2 g were selected and randomly divided into four groups ($n$ = 5 per group). The experimental groups were orally administered 200 µL of the screened strains suspension (1 × $10^8$ CFU/mL) at regular intervals for 21 days, whereas the control group received PBS. After a 7-day observation period, all mice were regularly monitored for clinical signs, and their body weights were recorded. At the end of the experiment, all mice were euthanized by cervical dislocation in accordance with

the updated version of the American Veterinary Medical Association (AVMA) Guidelines for the Euthanasia of Animals, to assess any pathological changes in their organs.

## Design of animal experiments

Considering probiotic characteristics and degradation rate, strain 38352 was used to participate in animal tests. A total of 210 healthy male broilers (1 day of age) were selected and randomly divided into six groups ($n = 30$ per group), including Control, Infection, Vaccine, OTA, Vaccine + OTA, and Vaccine + OTA + 38352 groups. All groups received the same basal diet and were housed under identical conditions with standardized equipment. Chickens were fed three times daily (8:00, 13:00, 18:00) under controlled temperature and humidity. A supplemental lighting system provided 16 h of light per day. Housing was regularly disinfected following standard protocols throughout the 28-day experiment. Group differences are shown in the following: (i) Control group, the chicks were given normal diet and water; (ii) Vaccine group (vaccine immunization control group), chicks were given 50 µg of vaccine in the neck at 5 days of age, and infected by *C. perfringens* 1 mL ($1 \times 10^8$ CFU/mL) at 19, 20, and 21 days of age; (iii) Infection group, chicks were gavaged with *C. perfringens* 1 mL ($1 \times 10^8$ CFU/mL) at 19, 20, and 21 days of age, once in the morning and evening; (iv) OTA group, chicks were given OTA by gavage at 10:00 am at 2, 3, and 4 days of age; (v) Vaccine + OTA group, chicks were injected 50 µg of vaccine in the neck at 5 days of age, gavaged with OTA at 2, 3, and 4 days of age, and infected by *C. perfringens* 1 mL ($1 \times 10^8$ CFU/mL) at 19, 20, and 21 days of age; and (vi) Vaccine + OTA + 38352 group, chicks were given normal immunization and infection, with daily OTA administration during the test, and gavaged OTA at 10:00 am at 2, 3, and 4 days of age, OTA-degrading strain 38352 ($1 \times 10^8$ CFU/mL) was mixed into the drinking water throughout the test period.

Blood samples were collected from each chick via the wing vein on 0, 7, and 14 days after immunization. After that, chicks were euthanized by carbon dioxide asphyxiation in accordance with the updated version of the AVMA Guidelines for the Euthanasia of Animals, and tissues were collected.

### Detection of specific IgG antibodies and toxin neutralization test

The laboratory previously purified *C. perfringens* type α toxin protein was used as the antigen for testing specific antibody levels in animal models. Specific antibody titer determination using the indirect ELISA endpoint method (28). Briefly, ELISA plates were coated with 500 ng/mL antigen at 4°C overnight, washed three times, and then blocked with 5% skim milk at 37°C for 2 h. Subsequently, diluted serum samples were added and incubated for 2 h, followed by three washes. Then, goat anti-chicken IgG-HRP antibody diluted at 1:8,000 was added and incubated for another 2 h. Finally, the TMB substrate solution was added and incubated in the dark at 37°C for 30 min.

Toxin neutralization test was performed according to the previously published methodology (29). *C. perfringens* α toxin was centrifuged, the supernatant and the prepared serum were mixed by 1:1 ratio and incubated at 37°C for 2 h. The blood plate was punched, and 10 µL intermixture of each hole was fostered at 37°C overnight to assess hemolysis. The ability to neutralize the toxin was judged by the size of the external hemolytic ring.

### Determination of tonsil-associated cytokines in the jejunum and cecum

According to the method of reference 30, the jejunal and cecal tonsil tissues of chickens were ground to extract RNA by Trizol Reagent (Invitrogen, Sigma-Aldrich, St. Louis, MO). The RNA was then reverse transcribed into cDNA using ReverTra Ace (TRT-101, TOYOBO, Shanghai, China), following the manufacturer's protocol. The cDNA was used for quantification and expression analysis with 2× S6 SYBR Green qPCR Mix (EnzyArtisan, Shanghai, China). Real-time quantitative PCRs were carried out using the Applied Biosystems 7500 Real-Time PCR System, following the manufacturer's instructions.

Relative expression of interleukin (IL)-1β, IL-4, IL-12, IL-17, and interferon gamma (IFN-γ) was quantified using the $2^{-\Delta\Delta CT}$ method with β-actin as the housekeeping control. The primers used in this study are listed in Table S1.

### Lesion score

After euthanasia, the pathological changes of each organ were observed and the intestinal damage was scored: the small intestine of each chicken was removed, the score range from 0 to 4, 0 = normal intestinal appearance; 0.5 = severe plasma and mesentery congestion; 1 = thin intestinal wall, fragile, small red spots; 2 = focal necrotic lesions; 3 = fairly large necrotic plaques, gas-filled intestine, and small blood spots; 4 = severe extensive necrosis, obvious bleeding, and large amount of gas in the intestine.

### Statistical analysis

Statistically significant differences between mean values of the parameters tested in the animal trial were analyzed with ANOVA (one-way ANOVA or two-way ANOVA) using GraphPad Prism 8 software. The differences were considered statistically significant if the $P$ values < 0.05 (*$P$ < 0.05, **$P$ < 0.01, ***$P$ < 0.001).

## RESULTS

### Three strains exhibited a certain degree of OTA degradation capability through isolation

Sixty-two gram-positive strains were isolated by the enrichment method from 28 traditional fermented food samples. They were inoculated in primary screening medium containing different OTA concentrations. At an OTA concentration of 50 µg/mL, three strains isolated from pickled cabbages exhibited visible growth, as indicated by turbidity in the liquid medium. These strains were designated as 38351, 38352, and 38362 (Fig. 1A through C). The degradation rate of 38352 was significantly higher than that of strains 38351 and 38362, or 51.4% of OTA (Fig. 1D).

### Three isolated strains were identified through genetic sequencing as belonging to *Lysinibacillus macroides* and *Priestia aryabhattai* species

The nearly complete sequences of the 16S rRNA gene and *gyrB* gene from strains 38351, 38352, and 38362 were cloned and analyzed. The constructed phylogenetic tree showed that 38362 and 38352 were identified as *Lysinibacillus macroides,* and 38351 as *Priestia aryabhattai* (Fig. 2).

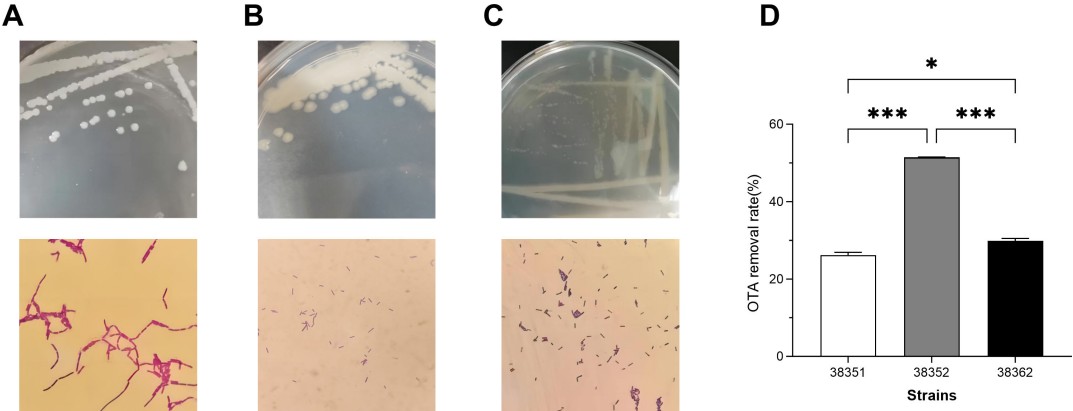

**FIG 1**  Isolation of the OTA-degrading strains. Colony and cell morphology of the isolated strains. (A) Strain 38351. (B) Strain 38352. (C) Strain 38362. (D) Removal ratio of OTA by strains. *$P$ < 0.05, ***$P$ < 0.001.

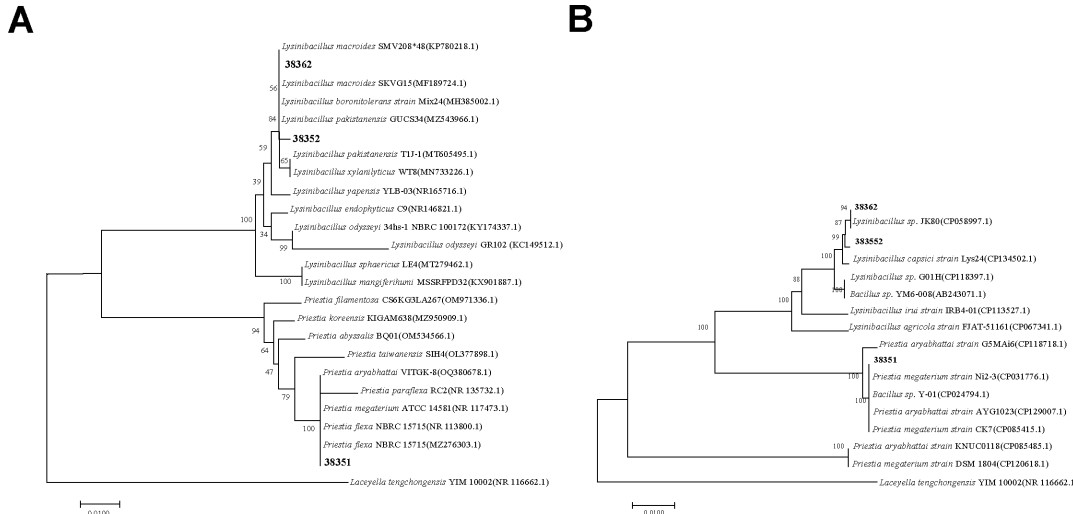

**FIG 2** Phylogenetic tree of the isolated 38351, 38352, and 38362 and related taxa. (A) Phylogenetic tree analysis of isolates based on partial 16S rRNA gene sequences. (B) Phylogenetic tree analysis of isolates based on partial *gyrB* gene sequences.

The 16S rRNA sequence had been uploaded to the NCBI GenBank and was obtained with the accession numbers OM232483.1 (38352), KY317957.1 (38362), and MN179986.1 (38351).

## All three OTA-degrading strains exhibited favorable probiotic characteristics

In the hydrochloric acid, sodium hydroxide, and bile salt tolerance tests, all three strains exhibited normal growth in LB medium. This indicates that these strains can withstand extreme conditions, including pH levels of 2 and 12.0 in hydrochloric acid, as well as 0.2% bile salt concentration, demonstrating their ability to adapt to harsh environments while maintaining bacterial activity (Table S2). Additionally, observations of the isolated colonies on the blood plate after 1 week revealed the absence of hemolytic rings, indicating that none of the three isolates demonstrated hemolytic activity (Table S2). Detailed results of the drug susceptibility test are presented in Table 1.

**TABLE 1** Results of bacterial susceptibility test[a]

| Strain | FFC (30 µg) | CZO (30 µg) | SH (100 µg) | MNO (30 µg) | TGC (15 µg) | MID (30 µg) | LZD (30 µg) | DOX (30 µg) | SM (10 µg) | DA (2 µg) |
|---|---|---|---|---|---|---|---|---|---|---|
| 38351 | S | S | S | S | S | S | S | S | S | R |
| 38352 | S | I | I | S | S | S | S | S | S | S |
| 38362 | S | R | R | S | S | I | S | S | S | I |

| Strain | AMP (10 µg) | VAN (30 µg) | AMX (20 µg) | TOB (10 µg) | CIP (5 µg) | RFP (5 µg) | OX (5 µg) | NAL (30µg) | POL (30 µg) | TLC (85 µg) |
|---|---|---|---|---|---|---|---|---|---|---|
| 38351 | S | S | R | S | S | S | S | S | R | S |
| 38352 | R | S | S | S | R | S | S | S | I | S |
| 38362 | R | S | I | S | R | S | S | S | S | S |

[a]FFC: florfenicol (30 µg), CZO: cefazoline (30 µg), SH: spectinomycin (100 µg), MNO: minocycline (30 µg), TGC: tigecycline (15 µg), MID: midecamycin (30 µg), LZD: Linezolid (30 µg), DOX: doxycycline (30 µg), SM: streptomycin (10 µg), DA: clindamycin (2 µg), AMP: ampicillin (10 µg), VAN: vancomycin (30 µg), AMX: amoxicillin (20 µg), TOB: tobramycin (10 µg), CIP: ciprofloxacin (5 µg), RFP: rifampicin (5 µg), OX: oxacillin (5 µg), NAL: nalidixic acid (30 µg), POL: fosfomycin (30 µg), TLC: ticarcillin (85 µg). The antibiotic susceptibility was performed according to the manufacturer and the Clinical and Laboratory Standards Institute guidelines (CLSI M02-A13 and M100-ED32). Inhibition zone diameters are interpreted as follows: <8 mm = resistant (R), 9–15 mm = intermediate (I), and >16 mm = susceptible (S).

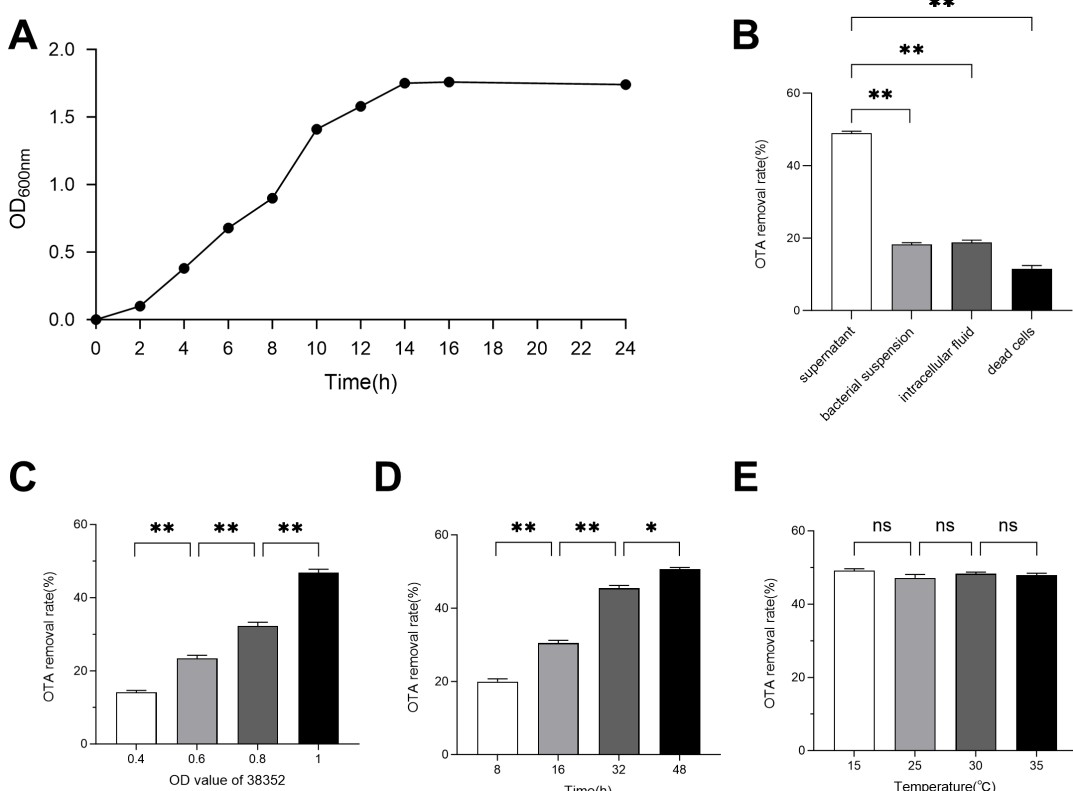

**FIG 3** Degradation properties of OTA by isolated strains. (A) Growth curve of isolated strain 38352. (B) Degradation rate of OTA by different cell components. (C to E) OTA degradation ability of strain 38352: (C) effect of different cell concentrations, (D) effect of culture time, and (E) culture temperature. *$P < 0.05$; **$P < 0.01$; ns, not significant.

## Strain 38352 demonstrated stable functionality in degrading OTA

### Growth curves of 38352

Combining the above test results, 38352 with good probiotics characteristics and the strongest degradation capacity was selected for downstream experiments.

To comprehensively study the growth characteristics of the isolated strain under different conditions, including temperature, pH, and nutrient availability, and to optimize its cultivation conditions, the growth curve of strain 38352 was analyzed. The growth curve of 38352 is shown in Fig. 3A. 38352 was in an upward trend before 14 h and entered a stable period after 14 h.

### Strain 38352 exhibits the most significant OTA removal activity in its culture supernatant

The OTA degradation mechanism of strain 38352 was verified by extracting culture supernatant, intracellular components, bacterial suspension, and dead cells. As shown in Fig. 3B, the degradation rates of OTA varied significantly among the four cellular components. The fermentation supernatant had the highest degradation rate, reaching 49.7%, while bacterial suspension, intracellular fluid, and dead cells had relatively weak degradation rates of 18.8%, 19.3%, and 12.53%, respectively. It can be seen that the removal of OTA by 38352 resulted mainly from the reaction degradation of the supernatant. The degradation of OTA by isolated strains is not dependent on cell adsorption, but rather on biodegradation dominated by active substances produced and secreted into the cell metabolism.

## Strain 38352 shows time- and concentration-dependent OTA degradation

The effects of temperature, time, and cell concentration on the OTA degradation rate are seen in Fig. 3. As the cell concentration increases, the degradation rate of OTA also gradually increases (Fig. 3C). At $OD_{600} = 1$, the degradation rate reached a high level of 47.5%. Over time, the degradation rate of OTA rises slowly until stabilization. After 32 h of treatment, OTA degradation reached a high level of 51% (Fig. 3D). The incubation temperature had no significant effect on the OTA degradation rate (Fig. 3E).

## Strain 38352 showed no apparent pathogenic effects in mice

As shown in Fig. S1A, mice in the experimental groups exhibited weight gain trends comparable to those of the control group, with the strain 38352 group even showing the most rapid growth (Fig. S1A). No fatalities occurred during the experiment, and the experimental groups did not exhibit any notable changes in fur condition, skin texture, mucous membrane appearance, eye health, respiratory patterns, or behavioral traits. Furthermore, no symptoms such as tremors, twitching, salivation, diarrhea, lethargy, or coma were observed. At the end of the experimental period, all mice underwent a complete pathological examination through dissection. Critical organs, including the heart, liver, spleen, lung, and kidney, were meticulously examined, and no abnormalities were observed in any of the groups. Specifically, there were no signs of hemorrhage, inflammation, necrosis, abnormal discoloration, or alterations in tissue texture. As shown in Fig. S1B, which presents a representative mouse from both the strain 38352 group and the control group, no pathological changes were detected in organs.

## Strain 38352 significantly alleviated the toxicity of OTA in chickens

### Strain 38352 alleviated the growth-restricting effects of OTA on broiler chicken weight

The findings revealed a consistent upward trend in all experimental groups (Fig. 4A). Notably, the Vaccine + OTA + 38352 group exhibited the highest weight gain, surpassing the other groups significantly and consistently maintaining an upward trajectory. Conversely, the OTA group displayed the smallest weight increase, with a growth rate lower than that of the other groups.

### Strain 38352 exhibited a protective effect against C. perfringens in broiler chickens

Figure 4B represents the recorded percentages relative to the initial body weight after 7 days of toxin exposure. It is evident that OTA and infection groups exhibit the lowest growth rates, while all other groups show higher rates compared to these two groups. Among them, the Vaccine + OTA + 38352 group demonstrates the highest growth rates and is significantly higher than the OTA group and infection group ($P < 0.01$), consistently showing an upward trend throughout the experimental period. The growth rate of the Vaccine + OTA group surpasses that of the OTA group, and the weight gain of the Vaccine group in broilers is also higher than that of the infection group and the Vaccine + OTA group, with a significant difference compared to the infection group ($P < 0.05$).

### Strain 38352 alleviated the toxic symptoms of OTA in chickens

At the ages of 19, 20, and 21 days after infection with *C. perfringens*, the Infection group exhibited typical symptoms, including diarrhea, yellow-brown or dark-brown feces, foul odor, frequent and rapid wing flapping, and intestinal mucosal bleeding, especially noticeable on the evening of day 20. The Vaccine + OTA group showed similar symptoms, but they were milder and improved by the age of 21 days, with the symptoms disappearing. The Vaccine group and Vaccine + OTA + 38352 group of broilers maintained normal dietary intake, feces, and mental status without any observable clinical symptoms. It can

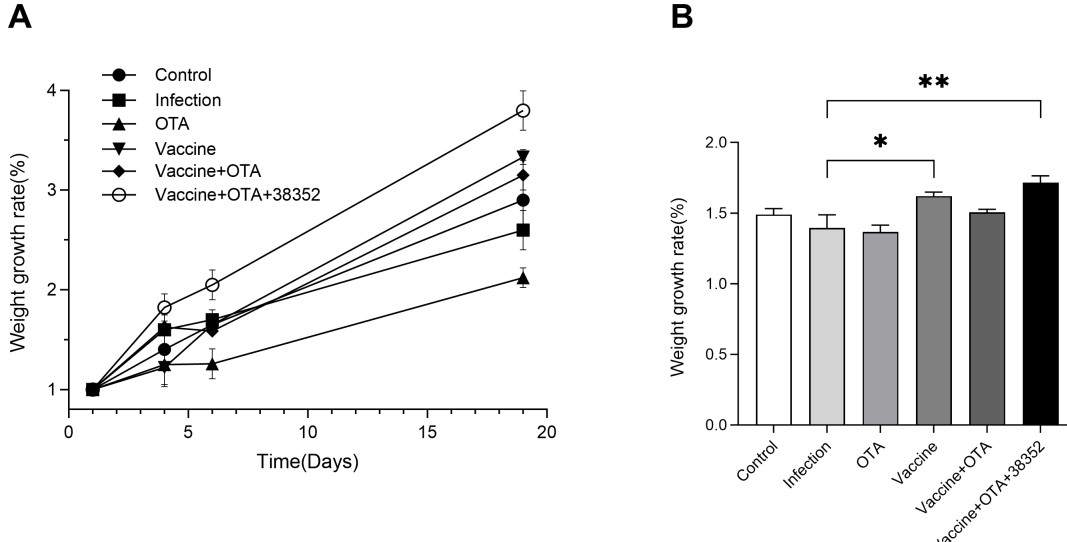

**FIG 4** Results of body weight change monitoring. (A) Line chart of body weight change before poison attack. (B) Histogram of body weight change after poisoning. *$P < 0.05$ and **$P < 0.01$.

be inferred that OTA had a certain inhibitory effect on the immune protection induced by the vaccine, while the isolated strain 38352 alleviated this inhibitory effect.

### Strain 38352 mitigated the immunosuppressive effects of OTA on vaccines

At 7 and 14 days after immunization, blood samples were collected to obtain serum from each group of chickens. The serum samples were then analyzed to measure the specific IgG antibody levels, as shown in Fig. 5. The Vaccine group and Vaccine + OTA + 38352 group had higher antibody titers compared to the Vaccine + OTA group. Specifically, on day 7, the overall antibody titer of the Vaccine group was higher than that of the Vaccine + OTA group, but the difference was not significant. However, the antibody levels in the Vaccine + OTA + 38352 group were extremely significantly higher than both the Vaccine group and the Vaccine + OTA group ($P < 0.01$). On day 14, the antibody titer in the Vaccine group was significantly higher than that in the Vaccine + OTA group ($P < 0.05$), and the antibody levels in the Vaccine + OTA + 38352 group were extremely significantly higher than both the Vaccine group and Vaccine + OTA group ($P < 0.01$).

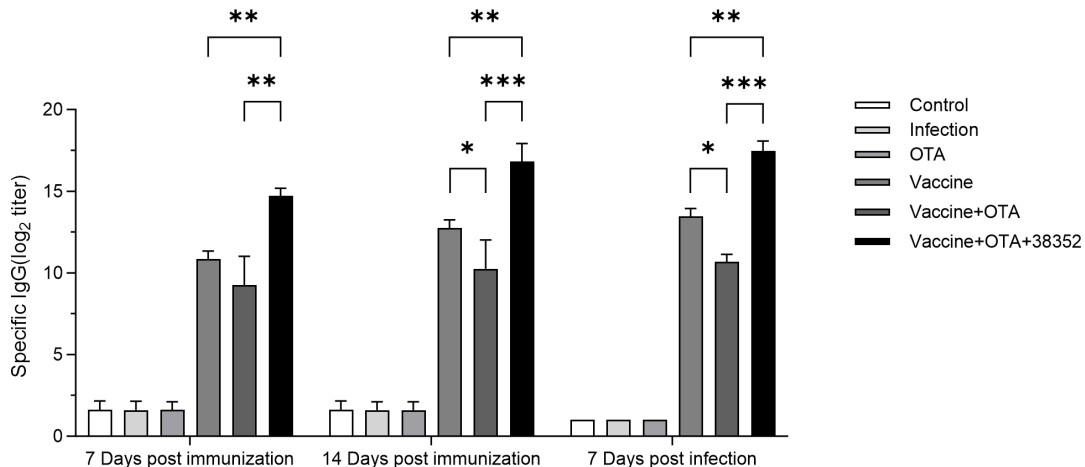

**FIG 5** Specific IgG antibody level in chicken serum on Day 7 and Day 14 after immunization, as well as 7 days after subsequent challenge with *C. perfringens*. *$P < 0.05$, **$P < 0.01$, and ***$P < 0.001$.

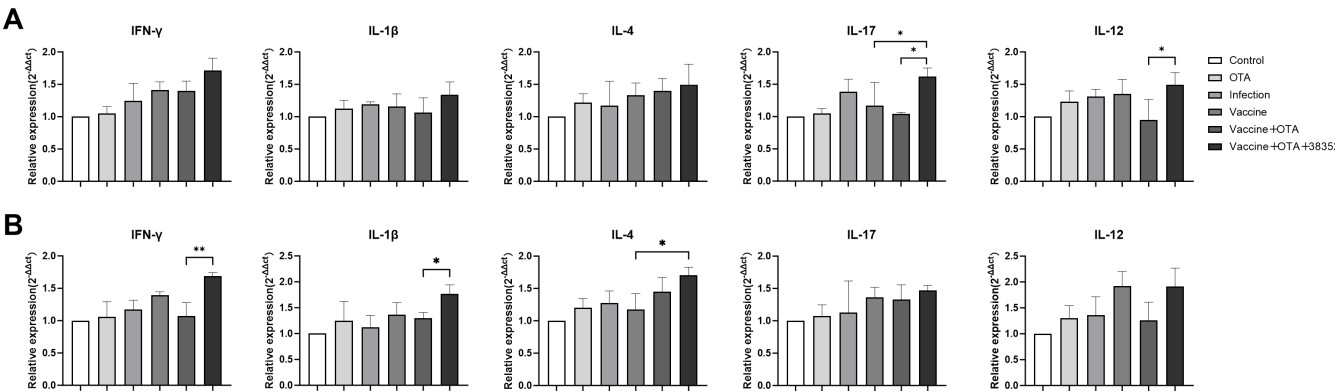

**FIG 6** Gene transcription of immune-related cytokines in (A) jejunum and (B) cecal tonsil. Data were presented as means ± SD (*$P < 0.05$ and **$P < 0.01$).

These results indicate that OTA can inhibit the production of specific antibodies induced by the vaccine to a certain extent. However, the positive controls, 38352 strains, can significantly ameliorate the immunosuppressive effects caused by OTA.

At 14 days after immunization, the chickens were infected with *C. perfringens*. Blood samples were collected 7 days after the challenge to obtain serum. The serum samples were then analyzed to measure the specific IgG antibody levels, as shown in Fig. 5. The antibody titers in the Vaccine group and Vaccine + OTA + 38352 group were all higher than those in the Vaccine + OTA group. Specifically, the antibody titer in the Vaccine group was significantly higher than that in the Vaccine + OTA group ($P < 0.05$). Moreover, the antibody levels in the Vaccine + OTA + 38352 group were extremely significantly higher than both the Vaccine group and the Vaccine + OTA group ($P < 0.01$).

The toxin neutralization assay results show that the OTA group exhibited a hemolysis zone diameter of 16.3 mm, slightly larger than the blank group (16 mm). The Vaccine + OTA group produced a hemolysis zone diameter of 15.5 mm, larger than the Vaccine group (15 mm). In contrast, the Vaccine + OTA + 38352 group exhibited the smallest hemolysis zone diameter, measuring 13.5 mm, significantly lower than the other groups (Table S3). This finding further supports the notion that 38352 can ameliorate OTA-induced immunosuppressive effects.

### Strain 38352 augmented the immunization levels of broiler chickens effectively

After 7 days of toxin exposure, broiler chickens were sacrificed, and samples of the jejunum and cecal tonsils were collected to evaluate mRNA expression levels of immune-related cytokines. The results, as depicted in Fig. 6A, revealed that compared to the control group, the Infection group exhibited higher mRNA expression levels of IL-1β, IL-12, and IL-17 genes compared to the Vaccine + OTA group. Moreover, the Vaccine + OTA + 38352 group showed elevated mRNA expression levels of IL-1β, IL-4, IL-12, IL-17, and IFN-γ genes compared to the other groups. Specifically, the IL-12 and IL-17 gene expression levels in the Vaccine + OTA + 38352 group were significantly higher than those in the Vaccine + OTA group ($P < 0.01$). Additionally, the Vaccine group displayed higher IL-12 gene expression levels than the Vaccine + OTA group. Furthermore, when compared to the Vaccine group, the Vaccine + OTA + 38352 group demonstrated a significant increase in IL-17 gene expression ($P < 0.01$). This suggests that oral administration of 38352 can enhance the mRNA expression levels of immune cytokines in broiler chickens.

The transcription levels of various genes in the cecal tonsils are depicted in Fig. 6B. Notably, the Vaccine + OTA + 38352 group showed higher gene expression levels compared to other immune groups. Specifically, when compared to the Vaccine group, the Vaccine + OTA + 38352 group exhibited a significant increase in IL-4 gene expression ($P < 0.05$). Additionally, compared to the Vaccine + OTA group, the Vaccine + OTA + 38352 group showed a markedly higher expression of IFN-γ ($P < 0.01$). These results indicate

that 38352 effectively enhances the Th1 and Th2 immune levels in broiler chickens. In comparison to the individual effects of the vaccine, the addition of 38352 significantly elevates the cellular and humoral immune levels in broiler chickens. The infection group of chickens exhibited a higher presence of virus-infected cells within their bodies compared to the blank group, which subsequently led to increased secretion of these cytokines, thereby strengthening the immune response. In summary, the incorporation of 38352 effectively enhances the Th1 and Th2 immune levels in broiler chickens, and their addition to the vaccine treatment significantly improves both cellular and humoral immune responses. The presence of a higher number of virus-infected cells in the infected group of chickens promotes the secretion of these cytokines, reinforcing the immune system.

### Strain 38352 effectively mitigated the development of intestinal lesions caused by C. perfringens infection.

In this experiment, significant clinical diseases were observed following the infection with *C. perfringens*. The broiler chickens that were not inoculated with Type A *C. perfringens* (Control group and OTA group) did not exhibit typical intestinal lesions. Compared to the infection group, all other vaccinated groups showed lower lesion scores. Specifically, the Vaccine group displayed a statistically significant difference in lesion scores compared to the Infection group. Moreover, the Vaccine + OTA + 38352 group exhibited significantly lower lesion scores compared to the Infection group. While the Vaccine + OTA + 38352 group showed a trend of reduced lesion scores compared to the Vaccine group, the differences were not statistically significant (Fig. S2). In summary, the experimental results indicate that the inoculation of broiler chickens with *C. perfringens* resulted in noticeable clinical diseases. In contrast, all vaccinated groups displayed lower lesion scores compared to the infection group. The Vaccine + OTA + 38352 group showed the most significant reduction in lesion scores, suggesting that the incorporation of 38352 can effectively mitigate the development of intestinal lesions caused by *C. perfringens* infection.

## DISCUSSION

OTA, a nephrotoxic and immunosuppressive mycotoxin, has long contaminated agricultural products worldwide, posing serious threats to food safety and livestock productivity (31, 32). Despite ongoing advances in microbial detoxification strategies, current approaches remain ineffective in addressing OTA-induced vaccine failure, a critical gap that exacerbates disease risks in intensive farming systems. Leveraging the microbial diversity of traditional fermented foods, this study hypothesized that multifunctional strains isolated from these sources could simultaneously achieve OTA detoxification and immune restoration. Through systematic screening of 28 types of traditional Chinese fermented foods, *L. macroides* 38352 was identified as a highly promising candidate strain. Experimental results fully supported this hypothesis: strain 38352 effectively degraded OTA via its culture supernatant and significantly increased vaccine-induced antibody titers in a broiler chicken model ($P < 0.01$). Notably, this study also revealed an unexpected finding: strain 38352 activated the jejunum and cecal tonsils through its metabolic products, significantly enhancing the Th1/Th2 cytokine network (significantly increased in IL-12 and IL-4), thereby promoting host immune defense in addition to detoxification. This dual-functional mechanism offers a novel "immune–detoxification synergy" paradigm for OTA control.

The three OTA-degrading strains isolated in this study exhibited desirable probiotic characteristics. Previous studies have shown that dietary supplementation with probiotics can enhance the detoxification of mycotoxins such as AFB1, thereby alleviating their negative impact on poultry productivity (33). Similarly, recent evidence suggests that probiotics may attenuate OTA toxicity by modulating host metabolic and immune pathways, including suppression of virulence-related gene clusters involved in iron acquisition and restoration of Nrf2/HO-1 pathway activity (34). Consistent with the

findings of Chen et al. (26), the OTA-degrading strains 38351, 38352, and 38362 isolated in this study were confirmed to possess desirable probiotic properties. To further assess their applicability as feed additives, we followed FAO/WHO guidelines to conduct toxicity assessments *in vivo* and performed genotypic and antibiotic susceptibility profiling (35). These strains demonstrated strong environmental resilience, maintaining viability across a broad range of pH values (2–12) and under elevated bile salt concentrations (0.2%), which suggests a high level of gastrointestinal tolerance. This adaptability may support their survival and functional activity in the host gut (36). Given that probiotics generally exert their effects by enhancing intestinal barrier integrity and modulating the gut microbiota, rather than directly eliminating pathogens (37, 38), the observed performance of these strains may be partly attributed to these mechanisms. While further studies are needed to elucidate the precise molecular interactions involved, our findings imply that these strains could serve as promising candidates for mitigating OTA toxicity while supporting intestinal health in livestock systems. Probiotics primarily exert their beneficial effects by improving the gut environment and enhancing intestinal barrier function, rather than directly eliminating pathogens.

Strain 38352 underwent comprehensive *in vitro* and *in vivo* safety assessments, demonstrating key probiotic safety traits, including broad-spectrum antibiotic sensitivity and a lack of hemolytic activity, both of which reduce barriers to potential industrial application. Huys et al. reported that approximately 30% of commercially available probiotic strains carry antibiotic resistance genes (39). In contrast, strain 38352 was sensitive to all 15 tested clinical antibiotics, suggesting a favorable safety profile compared to certain commercial strains, such as some *Lactobacillus* species (40). The observed antibiotic sensitivity of strain 38352 may be related to the absence of mobile genetic elements in its genome, which could limit the risk of horizontal gene transfer, although this assumption requires confirmation through whole-genome sequencing. Furthermore, a 21-day oral administration study in mice revealed no abnormalities in body weight or organ histopathology, supporting its non-toxicity under experimental conditions. Collectively, these results establish a preliminary safety framework for detoxifying probiotics and indicate that strain 38352 could be integrated into premix feed formulations following regulatory approval.

The culture supernatant of strain 38352 appears to play a central role in OTA removal, rather than the bacterial cells or intracellular components. Microbial detoxification of OTA is generally thought to proceed via two main mechanisms: (i) enzymatic degradation, in which extracellular enzymes hydrolyze OTA into less toxic products such as OTα and phenylalanine, and (ii) adsorption, where OTA binds non-covalently to peptidoglycan components of the cell wall (14, 41). For example, Sangare et al. found that the culture supernatant of the soil-derived bacterial isolate N17-1 was significantly more effective at degrading AFB1 than either live cells or intracellular extracts (42). In addition, OTA was also degraded by the cell-free supernatant of *Bacillus subtilis* CW 14 (43). Decarboxylase-mediated oxalic acid metabolism is important to antioxidation and detoxification in *Magnaporthe oryzae* (44). To elucidate the dominant mechanism in our study, four parallel treatments were conducted using culture supernatant, live bacterial suspension, intracellular extract, and heat-inactivated cells, all incubated with OTA (1 µg/mL) for 48 h under identical conditions. The results showed that the supernatant exhibited a markedly higher OTA degradation rate than the other three treatments, suggesting a primary role for secreted enzymes or metabolites. This observation is consistent with previous reports highlighting the significance of extracellular detoxifying agents in mycotoxin biotransformation (27). While further biochemical identification is needed, these findings indicate that certain active components present in the culture supernatant, likely proteins or enzymes, may be responsible for the observed OTA degradation, rather than the bacterial biomass or intracellular compounds. Notably, the relevance of secreted metabolic activity to toxin modulation is also supported by previous work on the Set2 family, which has been shown to regulate secondary metabolism and virulence in fungi (45). Together, these insights suggest that microbial

supernatants may represent a broadly applicable resource for mycotoxin mitigation strategies, although further studies are warranted to validate their mechanisms and effectiveness across different conditions.

Oral supplementation of Strain 38352 via drinking water was found to significantly attenuate OTA-induced immunosuppression and multi-organ pathological alterations in broiler chickens. OTA exerts its cytotoxic effects in a dose-dependent manner (46), impacting a range of intracellular processes, including oxidative stress, signaling transduction, and inflammatory responses. Both humoral and cellular immune responses are susceptible to OTA toxicity, with reported outcomes such as reduced immunoglobulin G (IgG) synthesis, decreased CD4$^+$ T-cell populations, and disruptions in cytokine secretion patterns. To investigate the potential immunomodulatory effects of strain 38352, this study quantified specific serum IgG antibody levels, as well as the concentrations of key cytokines, IL-2 and IFN-γ, in cecal and ileal tonsils following OTA exposure. Compared with both the vaccinated control and the OTA-challenged vaccinated group, broilers receiving strain 38352 displayed significantly elevated IgG levels ($P <$ 0.01), regardless of whether the probiotic was administered before or 7 days after OTA exposure. These results suggest that strain 38352 may enhance host immune responses under OTA-induced stress, potentially by modulating mucosal immune signaling or restoring cytokine balance. Overall, this study provides supportive evidence for the application of strain 38352 as a dietary strategy to counteract OTA-related immunotoxicity, offering new insights into the development of functional feed additives for improving animal health and food safety in contaminated production systems.

Strain 38352 significantly alleviated the inhibitory effects of OTA on body weight gain in broiler chickens. In poultry, OTA is known to induce nephrotoxicity and hepatotoxicity primarily by inhibiting the synthesis of proteins, DNA, and RNA, and by promoting lipid peroxidation, thereby impairing overall health and growth performance (47). Similar outcomes have been reported in other studies, such as those by Markowiak et al. (48) and Zhang et al. (49), which investigated the growth-promoting effects of synbiotics containing lactic acid bacteria, *S. cerevisiae*, or selenium-enriched yeast in OTA-challenged broilers. In our study, broilers in the vaccine + OTA + 38352 group achieved the greatest body weight gains among all experimental groups and maintained a consistently upward growth trajectory throughout the trial. In contrast, the OTA-only group exhibited the lowest weight gain, consistent with previously reported OTA-induced growth suppression. These findings suggest that *L. macroides* 38352 may not only mitigate the toxic effects of OTA but also contribute to improved growth performance under OTA stress conditions. The dual efficacy in detoxification and growth recovery observed in this *Bacillus* strain highlights its potential value in the development of multifunctional feed additives.

Strain 38352 effectively enhanced both Th1- and Th2-type immune responses in broiler chickens by correcting cytokine imbalances induced by OTA exposure. Previous studies have demonstrated that OTA exacerbates inflammation by suppressing anti-inflammatory cytokines such as IL-10 and IL-4 while upregulating pro-inflammatory mediators including TNF-α and IL-6 (50). Moreover, OTA impairs both humoral and cell-mediated immunity, further compromising host defense mechanisms (47). Cytokines play central roles in coordinating innate and adaptive immune responses to pathogens (51). While elevated levels of cytokines such as IL-1β, IL-12, IFN-γ, and IL-4 can indicate a heightened inflammatory state, moderate increases are often associated with effective immune regulation. IL-17, in particular, serves dual roles: it promotes the recruitment of immune cells to the site of inflammation by inducing pro-inflammatory chemokines (52) and contributes to the body's recovery by upregulating tissue repair-related factors (53). In our study, the mRNA expression levels of IL-1β, IL-4, IL-12, IL-17, and IFN-γ in the cecal and ileal tonsils were significantly elevated ($P < 0.01$) across all experimental groups on day seven post-OTA exposure, indicating activation of both cellular and humoral immune responses. Notably, the vaccine + OTA + 38352 group exhibited higher cytokine gene expression than the vaccine + OTA group, suggesting that supplementation with strain

38352 helped mitigate OTA toxicity and enhance the immunogenicity of the vaccine. In addition to immunological assessments, the vaccine + OTA + 38352 group also showed a higher weight gain rate compared to both the vaccine-only and vaccine + OTA groups. This finding implies a potential link between improved immune function and enhanced growth performance. Taken together, these results support the hypothesis that strain 38352 contributes to immune modulation under OTA-induced stress, thereby providing a mechanistic rationale for its use as a probiotic-based OTA detoxifying agent.

Strain 38352 exhibited a synergistic effect with vaccine antigens, enhancing protection against *C. perfringens* infection. Necrotic enteritis caused by *C. perfringens* remains a major concern in the poultry industry. Although research on aluminum-adjuvanted toxoid vaccines and recombinant protein-based formulations has progressed (54–56), no commercial vaccines are currently available, and further efforts are needed to develop effective immunization strategies. In this study, the use of a well-established NE model allowed us to assess both the protective effect of a CPMEA-OmpA2-BLP subunit vaccine alone and its combination with strain 38352. Compared to the vaccine-alone group, the combined treatment resulted in a more substantial reduction in intestinal lesions and higher levels of specific IgG antibodies, especially under OTA-immunosuppressive conditions. While Pan et al. reported similar enhancement of IgG levels using recombinant *Lactobacillus plantarum*, our study differs in two key aspects: (i) it uses a spore-forming *Bacillus* species, which offers better environmental robustness and (ii) it assesses immune modulation specifically in the presence of an OTA challenge, a factor rarely addressed in previous probiotic–vaccine synergy studies (57). Su Hyun Park et al. demonstrated that inducing effective *Actinobacillus pleuropneumoniae* (APP)-specific T-cell responses, particularly Th1, Th17, cytotoxic T lymphocytes (CTLs), and multifunctional T cells, enhances protective immunity against APP infection (58). One possible explanation for the enhanced efficacy is that it may help maintain intestinal immune homeostasis by restoring mucosal barrier function and local cytokine balance (58, 59), as evidenced by the reduced damage scores in both the ileal and cecal tissues. Given that the gastrointestinal tract is the first site of OTA exposure and the primary interface for immune activation, its integrity is critical for vaccine efficacy (47). The probiotics' effects may involve modulating host immune responses at the epithelial interface or influencing antigen presentation dynamics, though further molecular investigation is needed to clarify these pathways. On day 7 post-infection, intestinal lesion scoring revealed significantly reduced gut damage in both the vaccine-only and vaccine + 38352 groups, with the latter showing greater protective efficacy ($P < 0.01$). These findings suggest a potential strategy for enhancing subunit vaccine performance in OTA-contaminated environments by co-administering probiotics with immunomodulatory functions. Although these results are promising, their generalizability to commercial farming conditions and other pathogens remains to be tested. Nonetheless, the integration of detoxifying and immunoenhancing functions in a single probiotic strain may open new avenues for developing next-generation feed additives aimed at improving disease resistance and vaccine responsiveness in intensive livestock systems.

The majority of physicochemical methods are deemed impractical for OTA degradation, primarily due to incomplete toxin degradation, which restricts their application within the food industry or risks secondary contamination. In contrast, biologically mediated approaches do not compromise the nutritional value of food and exhibit high efficiency and specificity in detoxifying OTA, thus presenting promising prospects for widespread application (60, 61). The microbial strain employed in this study, *L. macroides* 38352, was originally isolated from Chinese fermented foods and has been demonstrated to possess favorable probiotic attributes and safety. Furthermore, the application of strain 38352 in animal models yielded significant outcomes, effectively alleviating OTA-induced immunosuppression, fostering the generation of specific antibodies, and enhancing overall organismal performance.

Despite revealing the considerable potential of strain 38352 in OTA detoxification and immune modulation, several limitations of this study warrant further investigation. First,

the mechanistic understanding of OTA degradation by strain 38352 remains incomplete. Critical parameters such as the minimum effective concentration required for degradation, the kinetics of the detoxification process, and the chemical identity and toxicity of OTA degradation products have not yet been fully characterized. Second, while our findings were demonstrated in a broiler chicken model, the efficacy and safety of this strain in other economically important species, such as swine or aquaculture animals, remain unknown. In addition, although short-term feeding trials indicated immunological and growth benefits, the long-term impacts of dietary supplementation with strain 38352 require further assessment. Specifically, its influence on carcass traits such as fat deposition and meat quality should be evaluated. Moreover, the molecular pathways through which strain-derived metabolites influence host energy metabolism remain unexplored. These limitations underscore the need for more comprehensive multi-species, long-duration, and mechanistic studies to substantiate the broader applicability of this probiotic candidate.

Future studies should clarify the molecular basis of OTA degradation, particularly the role of key enzymes such as amide hydrolases, and consider constructing engineered strains with enhanced efficiency. Whole-genome sequencing will allow us to better understand the genetic background of the strain. As demonstrated in the reference (62), whole-genome analysis should be prioritized in subsequent work to elucidate genetic features related to safety, adaptability, and metabolic potential, including the identification of antimicrobial resistance genes and functional loci involved in detoxification and immune modulation. Broader evaluations across livestock species and longer-term feeding trials are also needed to assess safety, metabolic impact, and practical applicability. Importantly, the observed synergy between strain 38352 and vaccine antigens suggests a new strategy for mitigating immunosuppression under mycotoxin stress. If validated further, this dual-function approach could inform the design of multifunctional feed additives for improved animal health and food safety. While this study offers only a modest step, it provides valuable insight into integrating detoxification and immune support through probiotic interventions in livestock production.

## Conclusion

In conclusion, *L. macroides* 38352 was successfully isolated from traditional Chinese fermented foods and demonstrated dual probiotic functions of efficient OTA detoxification and immune enhancement. The results demonstrate that strain 38352 possesses excellent environmental adaptability, maintaining viability under extreme pH conditions (2–12) and in the presence of 0.2% bile salts. Its safety was confirmed through comprehensive antibiotic susceptibility testing and acute oral toxicity assessment in mice. Furthermore, OTA degradation by strain 38352 was achieved primarily through extracellular supernatant activity, rather than traditional cell wall adsorption mechanisms. Dietary supplementation with this strain in OTA-contaminated feed not only improved broiler weight gain but also significantly increased vaccine-specific antibody titers. Notably, co-administration of strain 38352 with a *C. perfringens* vaccine reduced intestinal lesion scores and upregulated Th1/Th2 cytokine responses, suggesting a synergistic role in mucosal immune protection. Therefore, by introducing immune synergy as a core evaluation criterion, this study expands the framework of microbial detoxification research and provides an innovative strategy to address the dual challenges of OTA contamination and vaccine failure in intensive farming systems. These findings also underscore the potential of traditional fermented foods as underexplored sources of functional microbes. Future work should investigate the strain's efficacy under multi-mycotoxin stress and in diverse host species to advance microbiome-informed strategies that integrate detoxification and immune modulation for sustainable animal health management.

## ACKNOWLEDGMENTS

The authors would like to thank the National Natural Science Foundation of China, the State Key Laboratory of Veterinary Biotechnology Foundation, and Northeast Agricultural University for their support.

This research was supported by the earmarked fund for CARS-36 and the SIPT Project of Northeast Agricultural University (202310224043S, X202310224024, and X202310224172). This work is partly supported by the National Natural Science Foundation of China (Grant No. 31672532).

Jiaqing Wang: Writing—Review & Editing, Formal Analysis, Conceptualization. Haoyuan Duan: Writing—Original Draft, Data Curation, Visualization. Wenzhi Zhang: Methodology, Conceptualization, Investigation. Hai Li: Investigation, Methodology. Zaixing Yang: Methodology. Chuankun Zhang: Investigation. Shuhe Zhang: Data Curation. Junjie Guo: Investigation. Junwei Ge: Writing—Conceptualization, Project administration. Fang Wang: Writing—Conceptualization, Funding acquisition, Project administration. Mingchun Gao: Writing—Conceptualization, Writing—review & editing, Validation, Supervision, Funding acquisition. All authors read and approved the final manuscript.

The authors declare that they did not use AI and AI-assisted technologies in writing.

## AUTHOR AFFILIATIONS

[1]Heilongjiang Provincial Key Laboratory of Zoonosis, College of Veterinary Medicine, Northeast Agricultural University, Harbin, China

[2]State Key Laboratory of Animal Disease Control and Prevention, Harbin Veterinary Research Institute, Chinese Academy of Agricultural Sciences, Harbin, People's Republic of China

## AUTHOR ORCIDs

Jiaqing Wang  http://orcid.org/0009-0003-6274-059X
Junwei Ge  http://orcid.org/0000-0002-2460-1666
Mingchun Gao  http://orcid.org/0000-0001-8782-6080

## FUNDING

| Funder | Grant(s) | Author(s) |
| --- | --- | --- |
| The Earmarked Fund for CARS | CARS-36 | Mingchun Gao |
| The SIPT Project of Northeast Agricultural University | 202310224043S, X202310224024, X202310224172 | Junwei Ge |
| National Natural Science Foundation of China | 31672532 | Junwei Ge |

## AUTHOR CONTRIBUTIONS

Jiaqing Wang, Conceptualization, Formal analysis, Writing – review and editing | Haoyuan Duan, Data curation, Visualization, Writing – original draft | Wenzhi Zhang, Conceptualization, Investigation, Methodology | Hai Li, Investigation, Methodology | Zaixing Yang, Methodology | Chuankun Zhang, Investigation | Shuhe Zhang, Data curation | Junjie Guo, Investigation | Junwei Ge, Conceptualization, Project administration | Fang Wang, Conceptualization, Funding acquisition, Project administration | Mingchun Gao, Conceptualization, Funding acquisition, Supervision, Validation, Writing – review and editing

## DATA AVAILABILITY

The data that support the findings of this study are openly available in Mendeley Data at https://data.mendeley.com/datasets/fv84ndv3kb/7 (63). The 16S rRNA sequence had been uploaded to the NCBI GenBank and was obtained with the accession numbers OM232483.1 (38352), KY317957.1 (38362), and MN179986.1 (38351).

## ETHICS APPROVAL

The protocols and animal housing conditions were approved by the Animal Research Committee of Northeast Agriculture University (NEAUEC20210326). Furthermore, this study was conducted in compliance with the ARRIVE (Animal Research: Reporting of *In Vivo* Experiments) guidelines to ensure transparency, reproducibility, and ethical rigor in animal research reporting.

## ADDITIONAL FILES

The following material is available online.

### Supplemental Material

**Figure S1 (Spectrum02363-25-s0001.tif).** (A) Results of body weight change in mice; (B) results of mouse autopsy.
**Figure S2 (Spectrum02363-25-s0002.tif).** Lesion score results of each experimental group.
**Supplemental legends (Spectrum02363-25-s0003.docx).** Legends for supplemental material.
**Table S1 (Spectrum02363-25-s0004.docx).** PCR primer sequences.
**Table S2 (Spectrum02363-25-s0005.docx).** Probiotic properties of isolates.
**Table S3 (Spectrum02363-25-s0006.docx).** Diameter of hemolytic ring in each experimental group.

### Open Peer Review

**PEER REVIEW HISTORY (review-history.pdf).** An accounting of the reviewer comments and feedback.

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
