## [Reviewer comments · Microbiology Spectrum]

Microbiology Spectrum

***Lysinibacillus macroides* 38352 Isolated from Traditional Chinese Fermented Foods: A Dual Effect on Ochratoxin A Detoxification and Immune Suppression Alleviation**

Jiaqing Wang, Haoyuan Duan, Wenzhi Zhang, Hai Li, Zaixing Yang, Chuankun Zhang, Shuhe Zhang, Junjie Guo, Junwei Ge, Fang Wang, and Mingchun Gao

Corresponding Author(s): Mingchun Gao, Northeast Agricultural University

Review Timeline:

Submission Date:	July 31, 2025
Editorial Decision:	September 11, 2025
Revision Received:	September 27, 2025
Accepted:	October 28, 2025

Editor: Harold Marcotte

Reviewer(s): Disclosure of reviewer identity is with reference to reviewer comments included in decision letter(s). The following individuals involved in review of your submission have agreed to reveal their identity: Nurul Hawa Ahmad (Reviewer #1); Biao Tang (Reviewer #3)

Transaction Report:

DOI: <https://doi.org/10.1128/spectrum.02363-25>

Re: Spectrum02363-25 (*Lysinibacillus macroides* 38352 Isolated from Traditional Chinese Fermented Foods: A Dual Effect on Ochratoxin A Detoxification and Immune Suppression Alleviation)

Dear Ms. Jiaqing Wang:

Thank you for the privilege of reviewing your work. Below you will find my comments, instructions from the Spectrum editorial office, and the reviewer comments.

Revision Guidelines

Sincerely,
Harold Marcotte
Editor
Microbiology Spectrum

Reviewer #1 (Comments for the Author):

Overall, the manuscript was well-written. Research objectives and discussion were clearly mentioned. Nevertheless, methodology and results description require slight improvements as listed below:

L116-117: what does "different forms" refers to?

L145-147: The sentence is incomplete. Suggest to add "were analyzed?" at the end of the sentence
L152-153, 157: protocol on viable/live bacteria counts should be further described
L158-162: description of hemolytic experiment should be improved for overall clarity. Why there was a need to extend the experiment at 4C. Did you mean triplicates?
L166: Specify the volume of drug on each disc
Figure 1A - Missing strain label on each photo
Figure 1B - There was no statistical analysis indicated on the figure to support statement in the main text
Figure 3 - Similar comments as for Figure 1B
Are there any photos/images to support findings stated in section 3.5?
Figure 5A and 5B: It is more relevant to combine Day 7, Day 14-day after immunization, Day 14-immunized + infected with Cp in the same figure for comparison

Reviewer #2 (Comments for the Author):
Comments not provided.

Reviewer #3 (Comments for the Author):

This manuscript is well written, logically organized, and presents novel findings on the dual function of *Lysinibacillus macroides* 38352 in OTA detoxification and immune restoration. The study is timely and of high relevance to food safety and livestock health. I believe it has strong potential for publication after some revisions.

Major comments:

Since whole-genome sequencing has become cost-effective and rapid, it would be valuable to sequence this probiotic strain for more accurate identification, as well as to explore genomic features such as MLST, antimicrobial resistance genes, and potential OTA degradation-related genes.

To further enhance readability, the manuscript could be shortened slightly, especially in the "Results" section and Materials and Methods 2.8.

Minor comments:

Line 88: Please consider adding more references, e.g., DOI: 10.3389/fmicb.2022.823120.

Line 170: Sequencing and antimicrobial resistance gene analysis would strengthen the study; see DOI: 10.1128/spectrum.01257-22.

Line 124: Delete "(*C. perfringens*)".

Line 149: "3000" should be written as "3,000".

Line 168-169, 705: Please clarify the use of CLSI standards.

Line 174: The "*gyrB*" gene name should be italicized.

Spectrum02363-25

Title: *Lysinibacillus macroides* 38352 Isolated from Traditional Chinese Fermented Foods: A Dual Effect on Ochratoxin A Detoxification and Immune Suppression Alleviation.

Dear Dr. Harold Marcotte and the reviewers,

Thank you very much for the reviewing process of our manuscript entitled “*Lysinibacillus macroides* 38352 Isolated from Traditional Chinese Fermented Foods: A Dual Effect on Ochratoxin A Detoxification and Immune Suppression Alleviation.” (ID: Spectrum02363-25). We also highly appreciate the reviewer’s carefulness, conscientiousness, and beneficial suggestions. Those comments are all valuable and very helpful for revising and improving our paper, as well as guiding to our research. We have studied comments carefully and have made correction which we hope meet with approval. Revised portion are marked in red in the paper.

According to the Reviewer’s comments and instructions, we have made the following revisions:

Responses to Reviewer #1:

Comment 1: L116-117: what does "different forms" refers to?

Our reply:

Many thanks for the comments. The term “different forms” referred to the distinct colony morphologies observed on the LB agar plates, which was the primary criterion we used for the initial screening of suspected bacterial strains. Specifically, we selected colonies based on variations in key morphological characteristics such as size, shape (e.g., circular, irregular), color, surface texture (e.g., smooth, rough), elevation, and margin appearance.

The specific revisions are as follows:

“Colonies exhibiting distinct morphologies (based on size, shape, color, and margin) were selected as suspected strains and subjected to purification and preservation. The suspected strains with different forms were selected for purification and preservation.”

Please see Lines 114-116 of the revised manuscript.

Comment 2: L145-147: The sentence is incomplete. Suggest to add "were analyzed?" at the end of the sentence

Our reply:

Many thanks for the comments. We agree with the suggestion and have revised the sentence accordingly by adding “were analyzed” at the end as recommended. This modification has been implemented in the revised manuscript.

Please see Lines 144-146 of the revised manuscript.

Comment 3: L152-153, 157: protocol on viable/live bacteria counts should be further described

Our reply:

Many thanks for the comments. We agree that a more detailed description of the viable counting method is necessary for reproducibility. We have further described the protocol on viable/live bacteria counts to provide a comprehensive step-by-step protocol for the bile salt tolerance assay, including the viable count procedure. The specific revisions are as follows:

“Bile salt tolerance assay was performed according to the method described by Huinan Chen et al. with modifications (1). Briefly, the fresh bacterial culture was centrifuged at 3,000 rpm for 2 min. The supernatant was discarded, and the bacterial pellet was collected. The pellet was resuspended in LB liquid medium containing 0.2% (w/v) bile salts, thoroughly mixed by pipetting, and incubated at 37 °C for 2 h. After another centrifugation step to remove the supernatant, the bacteria were resuspended in an equal volume of sterile PBS buffer. To evaluate bacterial survivability, the number of viable cells was quantified. The resulting suspension was serially diluted 10-fold in sterile PBS. From each dilution, 10 µL was spotted onto LB agar plates, with three replicates per dilution. The plates were incubated at 37 °C for 24 h, and colonies were counted. The number of viable bacteria was expressed as colony-forming units per milliliter (CFU/mL).”

Please see Lines 153-162 of the revised manuscript.

Comment 4: L158-162: description of hemolytic experiment should be improved for overall clarity. Why there was a need to extend the experiment to 4 °C. Did you mean triplicates?

Our reply:

Many thanks for the comments. We have improved the description of the hemolytic experiment to enhance its overall clarity and reproducibility.

The phrase observation continued for a week at 4 °C was intended to describe a quality control measure where plates were stored at 4 °C post-incubation to allow for any delayed or subtle hemolytic reactions to become visible over time, and to ensure the stability of the observed results. Regarding the experimental replicates, we confirm that 'on three separate days' indeed indicates that the entire experiment was conducted three times independently, representing biological triplicates. The specific revisions are as follows:

“Hemolytic activity was determined on sheep blood agar plates (Biocell BioTech., Co., Ltd., Zhengzhou, China). Briefly, 200 µL of the bacterial culture was first spread onto LB agar plates and incubated at 37 °C for 24 h to obtain fresh colonies. Using a sterile pipette tip, a single colony was picked and gently spotted onto the surface of a sheep blood agar plate. The plates were incubated at 37 °C for 24 h and then examined for the presence of a clear zone (zone of hemolysis) around the bacterial growth. The diameter of the hemolytic zone was measured using a caliper. A positive control and a negative control were included in each assay. After that, the observation continued for a week at 4 °C. The entire experiment was independently conducted on three separate days to ensure reproducibility (biological triplicates).”

Please see Lines 163-171 of the revised manuscript.

Comment 5: L166: Specify the volume of drug on each disc

Our reply:

Many thanks for the comments. We have specified that the antibiotic susceptibility testing was performed using standard commercial discs (KONT Biology & Technology Co., Ltd., Wenzhou, China), with each disc containing a predefined volume of antibiotic solution as specified by the manufacturer. The detailed volume information for each antibiotic disc has been clearly provided

in Table 1 of the revised manuscript.

Please see Lines 172-174, 687-689 of the revised manuscript.

Comment 6: Figure 1A - Missing strain label on each photo

Our reply:

Many thanks for the comments. We have revised Figure 1A by adding clear labels (A, B, C) directly onto the photograph and have updated the figure caption accordingly to specify the strain identity for each panel.

The specific modifications are as follows:

“These strains were designated as 38351, 38352, and 38362 (Fig. 1A, 1B, 1C).”

Please see Lines 295-296 of the revised manuscript.

Comment 7: Figure 1B - There was no statistical analysis indicated on the figure to support statement in the main text. Figure 3 - Similar comments as for Figure 1B

Our reply:

Many thanks for the comments. We have carefully addressed the statistical concerns for both Figure 1B (revised Figure 1D) and Figure 3 and added the corresponding significance indicators (e.g., asterisks *) to support the statements in the main text. Additionally, we have updated the figure caption to specify the statistical test used and the significance levels (e.g., * $p < 0.05$, ** $p < 0.01$, *** $p < 0.001$). These revisions provide robust statistical support for our conclusions and have been incorporated into the revised manuscript.

Please see Lines 296 of the revised manuscript and the revised Figure 1D, and Lines 322-323 of the revised manuscript and the revised Figure 3.

Comment 8: Are there any photos/images to support findings stated in section 3.5?

Our reply:

Many thanks for the comments. We have added the following supporting materials and more detailed descriptions to further support the findings described in Section 3.5:

Figure S1A: Results of body weight change in mice.

Figure S1B: Results of mice autopsy.

The specific revisions are as follows:

“As shown in Fig. S1A, mice in the experimental groups exhibited weight gain trends comparable to those of the control group, with the strain 38352 group even showing the most rapid growth (Fig. S1A). No fatalities occurred during the experiment, and the experimental groups did not exhibit any notable changes in fur condition, skin texture, mucous membrane appearance, eye health, respiratory patterns, or behavioral traits. Furthermore, no symptoms such as tremors, twitching, salivation, diarrhea, lethargy, or coma were observed. At the end of the experimental period, all mice underwent a complete pathological examination through dissection. Critical organs, including the heart and liver, were meticulously examined, and no abnormalities were observed in any of the groups. Specifically, there were no signs of hemorrhage, inflammation, necrosis, abnormal discoloration, or alterations in tissue texture. As shown in Fig. S1B, which presents a representative mouse from both the strain 383526 group and the control group, no pathological changes were detected in organs.”

Please see Lines 336-347 of the revised manuscript.

Comment 9: Figure 5A and 5B: It is more relevant to combine Day 7, Day 14-day after immunization, Day 14-immunized + infected with Cp in the same figure for comparison.

Our reply:

Many thanks for the comments. We have reorganized the figure according to the suggestion. The data corresponding to Day 7 and Day 14 after immunization, as well as 7 days after subsequent challenge with *Clostridium perfringens* (Cp), have now been integrated into a single panel.

The specific revisions are as follows:

“Figure 5. The level of the specific IgG antibody. Specific IgG antibody level in chicken serum on Day 7 and Day 14 after immunization, as well as 7 days after subsequent challenge with *C. perfringens*.”

Please see Lines 374, 387 of the revised manuscript and the revised Figure 5.

Reviewer #2 (Comments for the Author):

Comments not provided.

Reviewer #3 (Comments for the Author):

This manuscript is well written, logically organized, and presents novel findings on the dual function of *Lysinibacillus macroides* 38352 in OTA detoxification and immune restoration. The study is timely and of high relevance to food safety and livestock health. I believe it has strong potential for publication after some revisions.

Major comments:

Since whole-genome sequencing has become cost-effective and rapid, it would be valuable to sequence this probiotic strain for more accurate identification, as well as to explore genomic features such as MLST, antimicrobial resistance genes, and potential OTA degradation-related genes.

To further enhance readability, the manuscript could be shortened slightly, especially in the "Results" section and Materials and Methods 2.8.

Our reply:

Many thanks for the comments. We have carefully reviewed the manuscript and have shortened the “Results” section as well as subsection 2.8 of “Materials and Methods” to improve conciseness and readability. Specifically, redundant descriptions and non-essential methodological details have been removed, while ensuring all key results and experimental procedures remain clearly communicated. We believe these revisions have enhanced the overall clarity and flow of the manuscript without compromising scientific rigor.

Minor comments:

Comment 1: Line 88: Please consider adding more references, e.g., DOI: 10.3389/fmicb.2022.823120.

Our reply:

Many thanks for the comments. We have added the suggested reference (DOI: 10.3389/fmicb.2022.823120) to the revised manuscript. This valuable citation further supports our methodological approach and strengthens the context of the discussion.

Please see Lines 87, 753-755 of the revised manuscript.

Comment 2: Line 170: Sequencing and antimicrobial resistance gene analysis would strengthen the study; see DOI: 10.1128/spectrum.01257-22.

Our reply:

Many thanks for the comments. We sincerely appreciate the reviewer's professional suggestion

regarding sequencing and antimicrobial resistance gene analysis, which undoubtedly holds significant value in strengthening the study. We have carefully reviewed the recommended reference (DOI: 10.1128/spectrum.01257-22) and gained meaningful insights from it.

However, due to current constraints related to time, budget, and technical expertise in whole-genome analysis, we intend to focus on this area and carry out related work in future studies. We have explicitly indicated in the revised manuscript that whole-genome sequencing should be conducted to further elucidate the genetic background of the strain. The specific modifications are as follows:

“Whole-genome sequencing will allow us to better understand the genetic background of the strain. As demonstrated in the reference (2), whole-genome analysis should be prioritized in subsequent work to elucidate genetic features related to safety, adaptability, and metabolic potential—including the identification of antimicrobial resistance genes and functional loci involved in detoxification and immune modulation.”

Please see Lines 623-627 of the revised manuscript.

Comment 3: Line 124: Delete “(*C. perfringens*)”. Line 149: "3000" should be written as "3,000".

Our reply:

Many thanks for the comments. We have carefully revised the manuscript accordingly. Specifically, we have deleted “(*C. perfringens*)” as suggested and corrected “3000” to “3,000”.

Please see Lines 123, 148 of the revised manuscript.

Comment 4: Line 168-169, 705: Please clarify the use of CLSI standards.

Our reply:

Many thanks for the comments. We have clarified the use of CLSI standards in the indicated lines of the revised manuscript. The specific modifications are as follows:

“The antibiotic susceptibility was performed according to the manufacturer and the Clinical and Laboratory Standards Institute guidelines (CLSI M02-A13 and M100-ED32).”

Please see Lines 179-180, 694-696 of the revised manuscript.

Comment 5: Line 174: The "gyrB" gene name should be italicized.

Our reply:

Many thanks for the comments. We have carefully reviewed the manuscript and corrected the formatting of the gene name "gyrB" to be italicized throughout the text.

Please see Line 185 of the revised manuscript.

We have gone through the manuscript very carefully, and made some improvements as suggested in the revised manuscript. These changes will not influence the content and framework of the paper. We deeply appreciate your consideration of our manuscript, and we look forward to receiving comments.

If we still have problems or faults, please tell us, we will try our best to revise. We appreciate for Editors/Reviewers’ warm work earnestly, and hope that the correction will meet with approval.

Once again, thank you very much for your comments and suggestions.

Yours sincerely

Mingchun Gao

E-mail: gaomingchun@neau.edu.cn

Reference

1. Chen H, Sun X, He H, Ren H, Duan H, Zhang C, Chang Q, Zhang R, Ge J. 2023. *Lysinibacillus capsici* 38,328 isolated from agricultural soils as a promising probiotic candidate for intestinal health. *Arch Microbiol* 205:251.
2. Tang B, Elbediwi M, Nambiar RB, Yang H, Lin J, Yue M. 2022. Genomic Characterization of Antimicrobial-Resistant *Salmonella enterica* in Duck, Chicken, and Pig Farms and Retail Markets in Eastern China. *Microbiol Spectr* 10:e0125722.

Re: Spectrum02363-25R1 (*Lysinibacillus macroides* 38352 Isolated from Traditional Chinese Fermented Foods: A Dual Effect on Ochratoxin A Detoxification and Immune Suppression Alleviation)

Dear Prof. Mingchun Gao:

Your manuscript has been accepted, and I am forwarding it to the ASM production staff for publication. Your paper will first be checked to make sure all elements meet the technical requirements. ASM staff will contact you if anything needs to be revised before copyediting and production can begin. Otherwise, you will be notified when your proofs are ready to be viewed.

Sincerely,
Harold Marcotte
Editor
Microbiology Spectrum

Reviewer #3 (Comments for the Author):

I think the author has addressed my concerns.